# Can Imaging Using Radiomics and Fat Fraction Analysis Detect Early Tissue Changes on Historical CT Scans in the Regions of the Pancreas Gland That Subsequently Develop Adenocarcinoma?

**DOI:** 10.3390/diagnostics13050941

**Published:** 2023-03-01

**Authors:** Ronald Lee Korn, Andre Burkett, Jeff Geschwind, Dominic Zygadlo, Taylor Brodie, Derek Cridebring, Daniel D. Von Hoff, Michael J. Demeure

**Affiliations:** 1Imaging Endpoints Research Laboratory, 7150 E Camelback Road Suite 120, Scottsdale, AZ 85251, USA; 2Hoag Family Cancer Institute, 1 Hoag Drive, Newport Beach, CA 92663, USA; 3Translational Genomics Research Institute, 445 N. Fifth Street, Phoenix, AZ 85004, USA

**Keywords:** pancreas cancer, radiomics, screening

## Abstract

Despite a growing number of effective therapeutic options for patients with pancreatic adenocarcinoma, the prognosis remains dismal mostly due to the late-stage presentation and spread of the cancer to other organs. Because a genomic analysis of pancreas tissue revealed that it may take years, if not decades, for pancreatic cancer to develop, we performed radiomics and fat fraction analysis on contrast-enhanced CT (CECT) scans of patients with historical scans showing no evidence of cancer but who subsequently went on to develop pancreas cancer years later, in an attempt to identify specific imaging features of the normal pancreas that may portend the subsequent development of the cancer. In this IRB-exempt, retrospective, single institution study, CECT chest, abdomen, and pelvis (CAP) scans of 22 patients who had evaluable historical imaging data were analyzed. The images from the “healthy” pancreas were obtained between 3.8 and 13.9 years before the diagnosis of pancreas cancer was established. Afterwards, the images were used to divide and draw seven regions of interest (ROIs) around the pancreas (uncinate, head, neck-genu, body (proximal, middle, and distal) and tail). Radiomic analysis on these pancreatic ROIs consisted of first order quantitative texture analysis features such as kurtosis, skewness, and fat quantification. Of all the variables tested, fat fraction in the pancreas tail (*p* = 0.029) and asymmetry of the histogram frequency curve (skewness) of pancreas tissue (*p* = 0.038) were identified as the most important imaging signatures for subsequent cancer development. Changes in the texture of the pancreas as measured on the CECT of patients who developed pancreas cancer years later could be identified, confirming the utility of radiomics-based imaging as a potential predictor of oncologic outcomes. Such findings may be potentially useful in the future to screen patients for pancreatic cancer, thereby helping detect pancreas cancer at an early stage and improving survival.

## 1. Introduction

Adenocarcinoma of the pancreas is one of the deadliest cancers, with an overall 5-year survival rate of 5–10%, because it is usually diagnosed at an advanced stage (80% of cases) when few therapeutic options are effective [1,2]. Due to its insidious growth, few or no symptoms occur until late in the disease or after it has metastasized to other organs [1,2,3]. Yet, prompt intervention at an early stage of the disease would yield far better outcomes and even the possibility of a cure [1,2,3]. Thus, early detection is critical. 

A key question regarding the dismal prognosis of patients with pancreatic cancer is whether it is the late diagnosis or the early dissemination of the disease to distant organs that is the primary cause of death. To understand the genetic evolution of pancreatic cancer, a quantitative genomic analysis was performed on multiple individuals who had died from advanced and widely metastatic pancreas cancer [4]. This study revealed that it took at least a decade for the cancer to develop in the pancreas, and another 5 to 7 years to acquire its metastatic potential before ultimately leading to the patient’s death another 2 years after that [4]. Such results not only contradict the idea that pancreatic cancers metastasize very early in their development, but more importantly, they open a broad window of opportunity to diagnose pancreas cancer early and potentially intervene promptly while the disease is still curable [4].

Due to the significant impact on patient survival of early pancreatic cancer detection, identifying subtle changes on imaging years before the cancer forms could lead to effective patient screening. Malignant tumors, including pancreas cancer, typically display substantial intra-tumor heterogeneity in virtually all phenotypic features, such as cellular morphology, gene expression (including the expression of cell surface markers, growth factor and hormonal receptors), metabolism, growth pattern including motility, desmoplastic stroma, as well as angiogenic, proliferative, immunogenic, and metastatic potential [5,6,7,8,9,10,11]. Imaging techniques based on radiomic analysis, heretofore referred to as quantitative texture analysis (QTA), can measure such intratumoral heterogeneity quantitatively, as was shown in several studies for different types of cancer including renal cell carcinoma, breast, lung, pancreas, and colorectal liver metastases where computed tomography (CT)-based texture features identified important prognostic factors linked to clinical outcomes [5,6,7,8,9,10,11]. These extracted QTA features can be used as inputs in machine learning algorithms to identify patterns and rules indiscernible to the naked eye. Once created and validated, such radiomic signatures can be used for the diagnosis and prognosis of cancer, and the longitudinal monitoring of tumor response after a therapeutic intervention [5,6,7,8,9,10,11]. 

Given the lag time in the development of pancreas cancer and the recent evidence regarding the utility of CT-based texture analysis as a prognostic tool for oncologic outcomes, we hypothesized that early changes in tissue texture within the normal pancreas of patients before they developed a readily visible pancreas cancer could be detected on contrast-enhanced CT (CECT) imaging, thereby revealing the signature of the pancreas cancer’s growth years before it actually developed. Therefore, the purpose of our study was to identify specific imaging features of the normal pancreas using QTA on CECT scans of patients who went on to develop pancreas cancer years later. 

## 2. Materials and Methods

The study was an IRB-exempt, retrospective, single institution study consisting of 22 patients (out of a total of 27 who underwent CT imaging) who had evaluable historical imaging data in the form of a CECT scan of the chest, abdomen, and pelvis (CAP) considered standard of care. The registry of patients from Hoag Memorial Hospital Presbyterian was queried for patients who had been diagnosed with pancreas cancer during the years 2015–2017 and who also had a CT scan of the abdomen and pelvis between 3.8 and 13.9 years prior. In total, 161 historical and 346 post-diagnosis CT scans from 27 patients were received, but as mentioned above, only 22 of the 27 patients had a full CECT historical imaging set (Portal Venous and/or Arterial) and a reference post-diagnosis imaging set that was used to identify the location of the pancreatic cancer.

The pancreas was divided into 7 regions (uncinate, head, neck-genu, body (proximal, middle, and distal), and tail) after which regions of interest (ROIs) were drawn on the normal pancreas gland (PG) obtained from these CECT images. A total of 154 unique pancreatic regions in these 22 patients were ultimately reviewed and analyzed. The imaging selected for analysis consisted of “healthy” pancreas tissue obtained between 3.8 and 13.9 years before the diagnosis of pancreas cancer was established. Although follow-up imaging was not evaluated, the earliest post-diagnosis scan for each subject was used to determine the future location of pancreatic cancer. No clinical information, prior read results, or any additional information were provided to the readers.

All scans were collected, processed, and controlled for completeness prior to measurement. Once a suitable imaging dataset was identified, it was transferred to TexRad (Essex, England) for QTA evaluation and Slice-O-Matic (Magog, QC, Canada) for fat fraction measurements. 

### 2.1. QTA Analysis of the Pancreas

A QTA evaluation of pancreas glands was measured at the single slice level in the axial view using TexRad. Due to the nature and angle of the imaging, the measured pancreas of each patient was drawn over the span of 1 to 3 slices to ensure full capture of all the pancreatic regions. Once the slice(s) with the most visible pancreatic regions was (were) identified, a 1–2 cm diameter ROI was drawn to capture the texture of the pancreas gland, as displayed in Figure 1 below. TexRad software contains a feature extraction algorithm that performs a pre-processing filtering step (Laplacian of the Gaussian) and a spatial scaling factor (SSF) that helps to calibrate radiologic images obtained on a variety of potentially different scanners, and acquisition has been described previously [12]. The resultant QTA outputs provide a histogram frequency curve of first order radiomic classifiers that allows for 6 different intensity-based features at each of 6 different SSF levels (0, 2, 3, 4, 5, and 6) for a total of 36 unique values per ROI. The intensity features include (1) mean pixel value, (2) standard deviation, (3) mean positive pixel value, (4) skewness, (5) kurtosis, and (6) entropy. 

### 2.2. Fat Quantification in Slice-O-Matic

The pancreas fat proportions in each region of the pancreas were calculated separately in Slice-O-Matic by redrawing the ROIs of each region of the pancreas using a (single vs. multiple slice) on the historical CECT. In this manner, the surface areas of each region that contained fat were divided by the total surface area of the regional ROI and the results were computed as the percent fat within a ROI. In this study a threshold pixel value range between −190 and 0 Hounsfield Units was used to define fat.

### 2.3. Statistical Analysis

Student *t*-tests were performed to determine whether significant differences existed for each of the measurements collected, between the group of regions that later developed pancreatic cancer and those that did not. These differences were taken at the region level. Therefore, although each patient in this study developed pancreatic cancer later in life, there were regions of each pancreas that did not develop disease and could be considered “healthy” tissue. A standard *p*-value of 0.05 was used to determine the significance of the separation between means. For each available measurement, a Receiver Operating Characteristic Area Under the Curve (ROC-AUC) was constructed to test the significance of said feature for binary classification purposes. Measurements that failed to achieve a reliable AUC (e.g., AUC ≥ 60% and *p*-value below the alpha level of 0.05) were converted from a continuous feature to a binary one by assigning a true or false value with respect to whether a subject’s measurement was above or below the population metric mean. Significant predictors were then selected as candidates for risk ratio analysis. The proportions of healthy tissue and those that later developed lesions were then extrapolated and compared against one another using the scikit-learn Python package [13]. Using these proportions, risk ratios were then constructed to highlight the associated impact of being in either group with respect to lesion development.

To assess the feasibility of using a signature compromised of QTA and Fat Quantification variables to predict lesion development, multiple logistic regression models were created and analyzed. A stepwise process of feature extraction was used to fit the top performing Ordinary Least Squares (OLS) Logistic Regression model. For each SSF level, all the significant features (as determined by the *T*-test and ROC-AUC analyses) were used as initial inputs for model creation. Individual features would then be manually pruned to determine if the significance and accuracy of the model improved and/or stabilized. Feature pruning would continue until each model began dramatically losing performance (e.g., Pseudo R-squared and the *p*-values for Log-Likelihood Ratio and individual features would increase). Once the feature-pruning process was complete for each SSF level, the overall top performing model was determined by ranking the Log-Likelihood Ratio (LLR) *p*-values and relative model accuracies.

## 3. Results

### 3.1. Patient Demographics

Our study population included 10 men and 12 women (mean age 79, range from 64 to 96). The average BMI for this population was 28.1 (range from 18.0 to 42.8). The 22 patients had stage 4 (*n* = 14), stage 2B (*n* = 2), stage 2A (*n* = 2), stage 1B (*n* = 1), stage 1A (*n* = 1), and stage 1 (*n* = 1) disease (one patient had no clinical stage). Most patients (*n* = 18) did not undergo surgical resection whereas four patients underwent surgical resection (two had a partial resection and two a complete resection). The historical CT scans were performed at a mean of 7.58 years prior to the CT that was used to diagnose pancreatic cancer (range 4.7 to 11.2 years).

### 3.2. Radiomic Analysis

Overall, the most comprehensive signature, as shown in Table 1, was obtained by applying a Spatial Scaling Factor (SSF) filter of 2 with the following features: mean pixel, skewness, whether a kurtosis value was above or below the population mean, and the proportion of pancreas fat observed.

The feature with the highest statistical significance was Mean Pixel value with a *p*-value of 0.005. Skewness and kurtosis mean split were also statistically significant, with *p*-values of 0.018 and 0.032, respectively. The *p*-value for total pancreas fat was above the 0.05 alpha threshold with a score of 0.172; therefore, it could not be deemed a reliable statistical feature when used by itself. Despite statistical insignificance, its inclusion improved model performance metrics such as Pseudo R-squared and LLR *p*-value and it did not break collinearity assumptions with the other features. Figure 2 below shows that this model achieved a receiver operating characteristic area under the curve (ROC-AUC) score of 0.7392 and an LLR-*p* value of 0.004.

The other main finding of the study was the discriminatory ability of the pancreatic fat to identify regions of the pancreas at risk of developing cancer. In particular, the proportion of fat in the tail region when compared to the rest of the pancreas was a predictor of future tumor development in that region of the pancreas. Indeed, patients with a tail fat percentage higher than 33% had a 75% chance of developing pancreas cancer, whereas patients with a lower tail fat percentage (i.e., less than the threshold of 33%) had a 9.1% chance of developing pancreas cancer. In other words, patients who contained more than the 33% fat threshold in the tail of the pancreas were 8.25 times more likely to develop pancreas cancer. Figure 3, below, displays the ROC-AUC of this threshold on the sample of measurable pancreas tails. It should be noted that unlike the QTA analysis results from 22 or 27 subjects with CECT scans available for analysis, the determination of regional pancreatic fat was possible in 26 or 27 subjects given that 4 additional subjects had non-contrast enhanced CT available fat fraction determination. 

Other individual parameters include skewness at SSF0. In particular, the skewness of any region (threshold value = −0.078) was a predictor of future tumor development in that region of the pancreas. Pancreatic regions with a skewness value above the threshold had a 33.3% chance of developing pancreas cancer, whereas those with a skewness value below the −0.078 threshold had a 13.6% chance of developing pancreas cancer. In other words, the pancreatic regions with skewness values higher than the threshold of −0.078 had a 2.46 risk ratio of developing pancreas cancer. Figure 4 displays the discriminative power of this relationship with an ROC curve.

## 4. Discussion

Our study showed that changes in texture in the pancreas, as measured on the CECT of patients who developed pancreas cancer years later, could be identified, thus confirming the potential utility of radiomics-based imaging as a potential predictor of oncologic outcomes. Although preliminary, the data demonstrated the feasibility of an imaging-based analysis to identify specific features within the pancreas that would lead to the development of cancer. In this manner, such methodology could be used as a potentially effective screening tool to identify populations at risk of developing pancreas cancer, thereby impacting clinical outcomes. As such, certain populations could be followed closely in order to offer prompt therapeutic intervention with curative intent as was recently demonstrated for high-risk individuals in the multicenter Cancer of Pancreas Screening Study (CAPS-5), where the 5-year survival of the patients with a screen-detected pancreas cancer was 73.3%, and median overall survival 9.8 years, compared with 1.5 years for patients diagnosed with pancreas cancer outside surveillance (hazard ratio (95% CI); 0.13 (0.03 to 0.50), P = 0.003) [14]. 

The study by Yachida et al. established for the first time that the timing of the genetic evolution of pancreas cancer was significantly longer than expected, where a decade or longer could occur between the initiating cellular mutation and the development of the parental, non-metastatic founder cell [4]. Then, a minimum of 5 more years would have to pass before cellular acquisition of metastatic ability and finally another 2 years before the patient affected by the cancer that had formed would succumb to the disease. This extremely slow progression from a single cellular mutation to a lethal cancer opened the door for intervention in the form of screening and therapy with curative intent [4]. One way to identify a population at risk of developing cancer is through imaging, as mammography has done for decades in patients with breast cancer and chest CT more recently with lung cancer. This long timeline to the development of pancreas cancer opens a window of opportunity to identify imaging features in the pancreas itself that could become the signature or “biomarker’ of a future tumor. 

Our hypothesis that distinct imaging features would arise in pancreatic tissue during the development of pancreas cancer was proven true as advanced texture analysis on CECT revealed such features. In particular, the asymmetry (skewness) of the histogram frequency curve (HFC) of pancreas tissue on CECT and fat fraction in the pancreas tail were identified as the most important imaging signatures of interest warranting further testing in future studies. Such findings are in keeping with those in glioblastomas where skewness and kurtosis accurately reflected the microscopic composition of treated glioblastomas [15]. This allowed the identification of pseudo-progression or early tumor progression, and as such provided invaluable information about treatment failure. In our study, the skewness of the HFCs obtained in the regions of the pancreas that subsequently develop pancreas cancer was significantly different from that obtained in the regions that did not develop pancreas cancer (*p* = 0.033), therefore confirming the utility of skewness as a reliable tool to screen patients for pancreas cancer. 

Fatty infiltration of the pancreas or “fatty pancreas”, which is synonymous with fatty degeneration in the pancreas, has recently been found to affect pancreatic insulin secretion and potentially act as a risk factor for pancreatic cancer. Changes in the fatty content of the pancreas at pathology along with fibrosis and inflammatory cell infiltration were found to be independent determinants of the development of pancreas cancer. Since imaging with CT can assess the fat content of the pancreas, it could, in theory, be used as a reliable predictor of the development of pancreas cancer. This is precisely what Fukuda et al. demonstrated where the fat content within the pancreas measured on CT was strongly associated with a fatty pancreas at pathology, itself subsequently independently associated with the development of pancreas cancer [16]. That study therefore established a potential role for imaging as a useful predictor of pancreas cancer [16]. Our results confirmed those of Fukuda whereby patients in our cohort who had a higher fraction of fat in their pancreas were more likely to develop pancreas cancer years later than those who did not (*p* = 0.029). Interestingly, this association was only present in the tail of the pancreas. Other regions of the pancreas did not show any meaningful correlation. Nevertheless, if fatty metamorphosis of the pancreas turns out to be a driver of pancreas cancer, then early intervention with lifestyle changes, newer GLP-1 agonist agents, and/or bariatric surgery in obese individuals may help reduce the risk of developing pancreas cancer. 

The main limitation of our study is its small sample size (*n* = 22 patients) and the lack of controls. It would have been useful to compare the texture of patients who developed pancreas cancer to those who did not. The data we collected are a first step towards a larger clinical trial where our findings could be appropriately tested. In that sense, our study is akin to a pilot clinical trial where a hypothesis about the utility of analyzing imaging texture as a potential screening tool for pancreas cancer was at the very minimum given some scientific credibility. Our findings will allow us and others to specifically test the veracity of using skewness and fat content within the pancreas for potential areas at risk of developing pancreas cancer. Another limitation is that the QTA analysis only provided first-order intensity-based classifiers. The use of first-order classifiers, however, seems justified as a pilot study as demonstrated by the importance of such features as skewness and kurtosis in other radiomic cancer studies. Nevertheless, other radiomic features (e.g., second- and higher order classifiers) may provide additional support to strengthen our model. We are currently evaluating such features on our advanced image analysis platforms. 

In summary, we found that certain imaging features in the pancreas of patients who would develop pancreas cancer years later were sufficiently different such that they could be considered predictive of the development of pancreas cancer. Our findings open the door to larger clinical trials that could ultimately lead to effective screening programs for patients deemed at risk of developing pancreas cancer. As a result, prompt therapeutic interventions on patients diagnosed early with pancreas cancer could dramatically alter survival outcomes. 

## Figures and Tables

**Figure 1 diagnostics-13-00941-f001:**
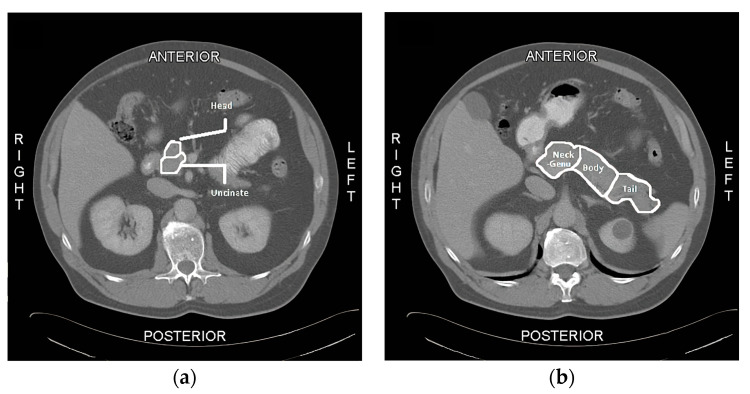
Regions of the Pancreas. This figure displays examples of the defined regions for the quantitative textural analysis (QTA) and fat quantification analysis of the pancreas: (**a**) the regions for the head and uncinate process of the pancreas; (**b**) the regions for the neck-genu, body, and tail of the pancreas. In this example, only 5 or the 7 ROIs are displayed.

**Figure 2 diagnostics-13-00941-f002:**
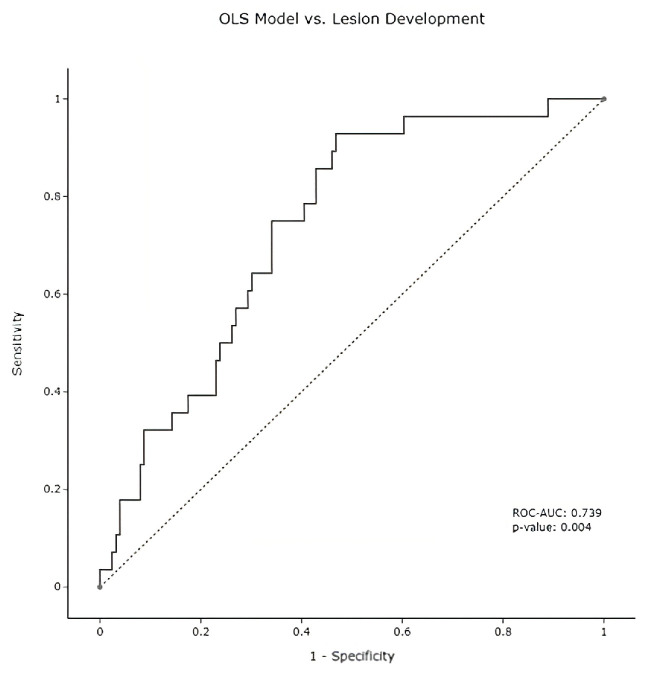
OLS ROC Curve. Receiver operating characteristic (ROC) curve in patients (*n* = 22) for the prediction of pancreas cancer development using the following baseline SSF2 radiomic features (mean, skewness, kurtosis mean split, and total pancreas fat %). This model was statistically significant with an Area Under the Curve (AUC) = 0.7392, *p*-value = 0.004.

**Figure 3 diagnostics-13-00941-f003:**
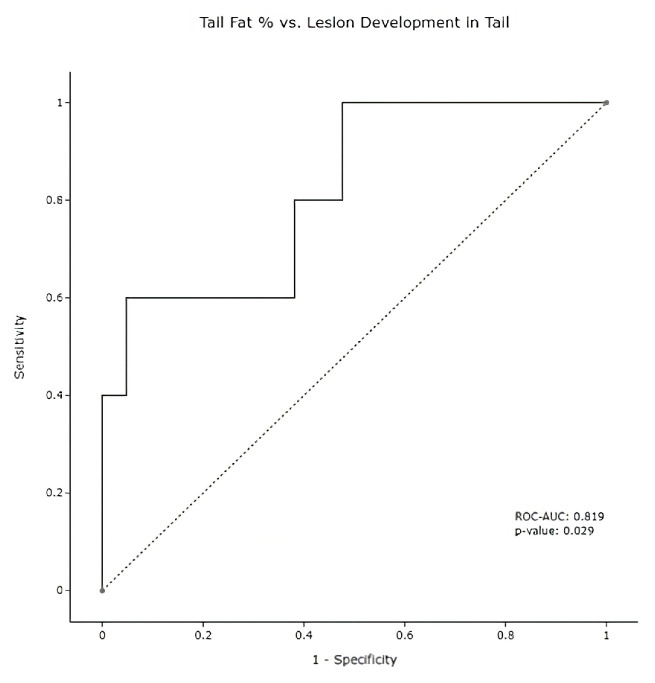
Pancreas Tail ROC Curve. Tail pancreatic fat measurements in subjects (*n* = 26) using only the pancreatic tail fat percentage as a feature; a classifier with an ROC-AUC of 0.819 was noted (*p* = 0.029).

**Figure 4 diagnostics-13-00941-f004:**
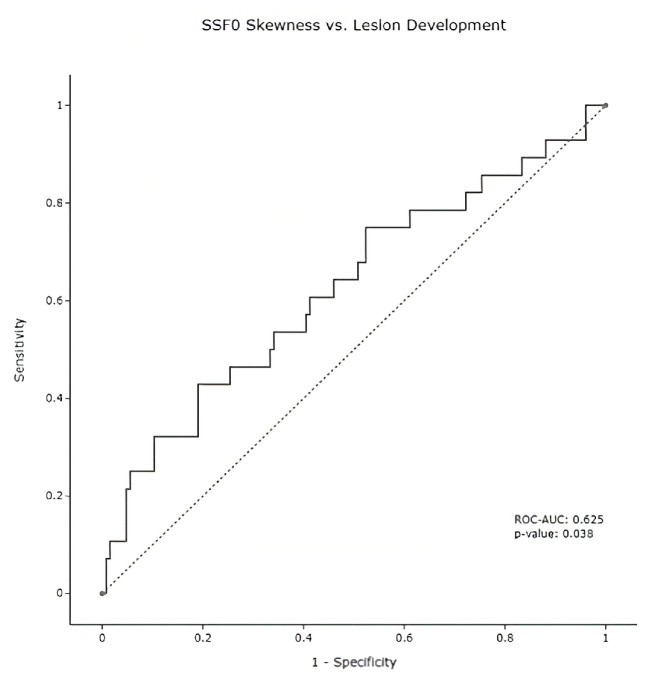
Lesion Skewness ROC Curve. Overall, SSF0 skewness was measurable on 154 pancreatic regions. Skewness measurements at baseline timepoints achieved a 0.625 ROC-AUC when discriminating for future lesion development, with a statistical significance of 0.038.

**Table 1 diagnostics-13-00941-t001:** Ordinary Least Squares (OLS) Logistic Regression Model.

Variable Name	Coefficient	Odds-Ratio	*p*-Value
Intercept	−1.0574	0.347	0.007
Mean	−0.0338	0.967	0.005
Skewness	1.0754	2.931	0.018
Kurtosis Mean Split	0.9913	2.695	0.032
Total Pancreas Fat %	−2.9476	0.052	0.172

Using the variables listed above, a logistic regression model was created to classify Lesion Development of pancreatic regions. In total, this model was calculated using 154 regions, achieved a Pseudo-R^2^ of 0.104, and retained statistical significance against the null model with a *p*-value of 0.004.

## Data Availability

Data will not be made available.

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
