# Peer review of "Can Imaging Using Radiomics and Fat Fraction Analysis Detect Early Tissue Changes on Historical CT Scans in the Regions of the Pancreas Gland That Subsequently Develop Adenocarcinoma?"

_diagnostics, 2023, doi:10.3390/diagnostics13050941_

Round 1

Reviewer 1 Report

My recommendations to the authors is that they should attach the anthropometric characteristics of the study population. To consider other covariates that may strengthen the development of pancreatic cancer.

Author Response

The average BMI along with the sample population range has been added to line 160 in an effort to describe anthropometric characteristics. Covariates were not directly considered in the interest of time.

Reviewer 2 Report

Main Comment:

This manuscript deals with imaging using radiomics and fat fraction analysis to detect early tissue changes on historical CT scans in the regions of the pancreas that subsequently develop adenocarcinoma. A retrospective evaluation of a small number of patients is presented. The authors label this as "akin to a pilot clinical trial […]", "preliminary" and "a first step towards a larger clinical trial". In this context this work can be seen as an incentive to further investigation.

Additional Comments/Suggestions:

Please check the numbers; e.g., Lines 160/161: "The 22 patients had stage 4 (n = 14), stage 2B (n = 2), stage 2A (n = 1), stage 1B (n =1), stage 1A (n = 1), and stage 1 (n = 1) disease (one patient had no clinical stage)." 14+2+1+1+1+1+1 is not 22.

Lines 111-113: "The resultant QTA outputs provides a histogram frequency curve of first-order radiomic classifiers that allows for 6 different intensity-based features at each of 6 different SSF levels (0,2,3,4,5 and 6) for a total of 36 unique values per ROI."-> The resultant QTA outputs provide a histogram frequency curve…

Lines 196-198: "In other words, the risk ratio of patients who contained more than the 33% fat threshold, in the tail of the pancreas, were 8.25 times more likely to develop pancreas cancer."-> In other words, patients who contained more than the 33% fat threshold…

Line 233: "Yashida et al." -> Yachida et al.

Lines 314-316: "Acknowledgments: In this section, you can acknowledge any support given which is not covered by the author contribution or funding sections. This may include administrative and technical support, or donations in kind (e.g., materials used for experiments)." – If there are no acknowledgments to be mentioned, this paragraph is to be deleted.

Author Response

Line 161 has been revised to change “Stage 2A (n=1)” to “Stage 2A (n=2)”. Line 112 has been revised to correct the grammar to “provide” from “provides”. Line 197 has been revised to remove “the risk ratio of” from the sentence. Line 233 has been revised to correct the spelling of “Yachida”. Line 310 has been revised to remove “generously” from the funding statement. Lines 314-316 have been removed since no Acknowledgements were required.